# Schizophrenic patients with type 2 diabetes: An 8-year population-based observational study

**Mariusz Jaworski**[1]*, **Mariusz Panczyk**[1], **Andrzej Śliwczyński**[2], **Melania Brzozowska**[2], **Joanna Gotlib**[1]

**1** Department of Education and Research in Health Sciences, Faculty of Health Sciences, Medical University of Warsaw, Warsaw, Poland, **2** Department of Analysis and Strategy, The National Health Fund, Warsaw, Poland

* mjaworski@wum.edu.pl

**Data Availability Statement:** All relevant data are within the manuscript and its Supporting Information files.

## Abstract

This paper presents a realistic evaluation of the prevalence of type 2 diabetes mellitus (T2DM) among Polish schizophrenic patients who have sought treatment through the Polish National Health Fund in the years 2010–2017. Data from the National Health Fund database was used and T2DM and schizophrenia groups were defined according to International Classification of Diseases (ICD-10) codes. Demographic data were collected from the web page of Statistics Poland (GUS). The annual prevalence of T2DM and schizophrenia was estimated, and the age groups were categorised into eight sets. The incidence of schizophrenia in T2DM patients in the years 2010–2017 was measured, including relative risk and 95% confidence interval (95% CI). The incidence of T2DM has been assessed in various subtypes of schizophrenia. In the eight years of follow-up study, 1,481,642 patients with schizophrenia were included, of which 185,205 were also diagnosed with T2DM. This accounted for 12.50% of all patients with schizophrenia. The trend of comorbid schizophrenia (F20) and T2DM (E11) in the general population of patients with schizophrenia, who sought treatment through the National Health Fund, was relatively stable in the years 2010–2017. The relative risk of T2DM in those with schizophrenia was 8.33 (95% CI 8.23–8.43) in 2017. Taking actions to enable the detection of diabetes in patients with concomitant schizophrenia is well-grounded, although these actions should be gender-dependent. There is also a need to take adequate actions to improve the efficiency of diabetological care among patients with schizophrenia.

## Introduction

Schizophrenia is a severe mental illness (SMI), characterised by cognitive, emotional, perceptual and behavioural disorders [1, 2]. The prevalence of schizophrenia is similar in many countries and the lifetime risk of schizophrenia is around 0.2–1% [1] with no difference between sexes. It is believed that schizophrenia is a disease that has serious consequences not only for one's health (disability, comorbidities, and complications), but also economic and social

**Funding:** The author(s) received no specific funding for this work.

**Competing interests:** The authors have declared that no competing interests exist.

sequelae. In addition, it is emphasised that patients with schizophrenia receive fewer healthcare services compared to the general population [3].

As a chronic disease, schizophrenia has been discussed as a potential risk factor for diabetes. This is due to the fact that some patients, newly diagnosed with schizophrenia, and who are not on antipsychotic drugs are also diagnosed with diabetes [4, 5]. Moreover, several genetic studies consider the genetic predisposition to diabetes in patients with schizophrenia. Their results, however, are unclear and require further research [4]. Additional research stresses that schizophrenia is associated with an increased risk for developing type 2 diabetes (T2DM). The more frequent prevalence of T2DM in patients with concomitant schizophrenia may be connected with schizophrenic symptoms (e.g. cognitive disturbances, and negative and positive symptoms) as well as the treatment used (e.g. the side effects of psychotropic medications), which can bear negative effects on the patient's lifestyle. This, in turn, translates into more frequent development of diabetes and complications thereof [5]. In this regard, the development of effective methods of medical treatment in patients with schizophrenia and concomitant T2DM, demands, in the first place, becoming familiar with data on the coexistence of the two illnesses, and in particular defining their trends, including not only age, but also sex. The presentation of such data will enable the development of adequate therapeutic treatment in patients with schizophrenia and T2DM. This is particularly critical, as the long-term outcomes of patients with both diabetes and schizophrenia remains unclear [6].

It is difficult to determine the exact frequency of the coexistence of schizophrenia and T2DM as the published results vary widely. For example, it is assumed that diabetes occurs two to five times more frequently in schizophrenia than in the general population [4]. A systematic review and meta-analysis conducted by Vancampfort et al. [7] indicated that the risk of T2DM in patients with schizophrenia was twice as high compared to healthy controls. Any large discrepancies may be related to methodological problems: for example, sample size estimation; methods of diagnosing T2DM and schizophrenia; the form of research (cross-sectional or longitudinal studies); and the criteria for inclusion in the study (such as age and sex). Therefore, it is important to select the appropriate sample size, which would take into account not only the appropriate number of patients, but also their age and sex characteristics [4]. For this reason, it may be helpful to analyse large databases that include the total number of patients receiving medical care. Such databases include patients with medical diagnoses from all over the country. By using such databases, it is possible to estimate the actual co-occurrence of T2DM and schizophrenia, thus, obtaining more realistic indicators, as well as minimising the risk of measurement error to a minimum.

This is important as the rate of co-occurrence of schizophrenia and T2DM cited in the literature is vague. Moreover, this indicator does not take into account the prevalence of T2DM in different subtypes of schizophrenia. This indicator treats schizophrenia as a uniform disease, while the International Classification of Diseases (ICD-10) codes for schizophrenia (F20) include paranoid schizophrenia, hebephrenic schizophrenia, catatonic schizophrenia, undifferentiated schizophrenia, post-schizophrenic depression, residual schizophrenia and simple schizophrenia. Therefore, it is crucial to verify if T2DM occurs with the same frequency in different subtypes of schizophrenia. Currently, it is difficult to determine which of the subtypes of schizophrenia may have a greater tendency to coexist with T2DM however this detailed data would allow for the development of more effective health education methods, and also provide more reliable and credible data overall.

Considering the scarcity of adequate studies, the aim of this paper was to evaluate the prevalence of T2DM among Polish schizophrenic patients who sought treatment through the Polish National Health Fund (Narodowy Fundusz Zdrowia, hereafter abbreviated to NFZ) in the years 2010–2017. The database developed by the NFZ includes all patients using public

healthcare in Poland. The coexistence of T2DM and schizophrenia in data from the years 2010–2017 was analysed without division into its subtypes in publications however it is important to bear in mind that schizophrenia is not a homogeneous illness, and special attention should be paid to all subtypes: paranoid schizophrenia (F20.0), hebephrenic schizophrenia (F20.1), catatonic schizophrenia (F20.2), undifferentiated schizophrenia (F20.3), post-schizophrenic depression (F20.4), residual schizophrenia (F20.5) and simple schizophrenia (F20.6) [8]. It is important to identify which subtypes of schizophrenia may bear more serious consequences in the context of treating T2DM. In this paper, we wanted to determine if T2DM is present in every subtype of schizophrenia however there is no such data available at the moment. In the context of all subtypes of schizophrenia, comparisons have been made regarding the frequency of their coexistence with T2DM among Polish patients, who sought treatment through the NFZ in the years 2010–2017, taking sex into account.

## Materials and methods

### Materials

**Definition of diabetes mellitus and schizophrenia.** Criteria for patient inclusion to the study were: 1) T2DM diagnosed according to the ICD-10 by a diabetologist [9]; 2) schizophrenia (F20) diagnosed by a psychiatrist according to ICD-10 [9]; and 3) patients benefiting from state medical care in the years 2010–2017. The necessary consents for data use were obtained from NFZ. All participants were diagnosed with T2DM according to the ICD-10 [9] at baseline, and all patients over the age of ten had a diagnostic code (ICD-10) of T2DM (E11), more than once in a given year from January 2010 to December 2017. All participants made at least one visit to outpatient or inpatient care relating to their T2DM diagnosis. We defined the T2DM group diagnosed with both T2DM and schizophrenia (F20) according to the ICD-10 codes at baseline [9].

On the basis of the inclusion criteria, from the entire database of patients with diabetes, only those fulfilling all the criteria mentioned were included. Data involving state medical services from 2010 to 2017 in the case of patients with comorbid T2DM and schizophrenia were taken into consideration. The aim of the study was to conduct a detailed analysis of the phenomenon discussed. In order to determine the current number of patients with T2DM, as well as the ratio of patients with comorbid T2DM and schizophrenia, in relation to the entire population with T2DM, data including patients benefiting from state medical care in the years 2010–2017 was analysed. The number of all subtypes of schizophrenia (from F20.0 to F20.9 [9]) was summed up for the overall evaluation of the prevalence of schizophrenia in T2DM. Three subtypes of schizophrenia were selected for detailed analysis: paranoid schizophrenia (F20.0); hebephrenic schizophrenia (F20.1); and residual schizophrenia (F20.5).

**Sample size.** The sample size included all patients registered in the NFZ database who used medical services financed from public funds [8]. This data is widely accepted to be representative of the entire Polish population, which validates its utility as a data source for population-based nationwide studies.

In the first step, for each year of observation, the number of patients with insulin-dependent diabetes indicated as the main cause of intervention (ICD-10 coding of E11; T2DM [9]) was generated from the NFZ database. Health declarations associated with each patient were analysed based on a unique patient identification ID (the PESEL number in Poland), which was anonymised prior to the analyses. It is now generally acknowledged that T2DM is diagnosed in adult patients, however, in the NFZ database, there are also T2DM diagnoses in children and adolescents (patients aged 10 to 18 years) according to ICD-10 [9]. On the basis of these medical diagnoses reported to the NFZ, these patients were included in the general group of

patients diagnosed with T2DM analysed in the present paper. All patients with diabetes included in the study benefited from state publicly funded medical care in the years 2010–2017. The majority of patients with diabetes benefit from state publicly funded medical care, where they are entitled to partial medicine reimbursement, specialised medical examinations and diabetics care without incurring additional fees. It should be noted that the treatment of diabetes and its complications is very expensive (e.g. hospitalisation, outpatient care, medications). Therefore, in the context of diabetes treatment in Poland, this group of patients enjoys the benefits of public rather than private funding [10].

In the second step, patients diagnosed with schizophrenia were selected. The subtype of schizophrenia was also included. In particular years of observation, the following total numbers of patients with schizophrenia were selected: 80,807 in 2010, 184,556 in 2011, 189,683 in 2012, 188,536 in 2013, 187,151 in 2014, 185,651 in 2015, 183,587 in 2016 and 181,716 in 2017.

In the final step, patients with coexisting T2DM and schizophrenia were selected, regardless of subtype. For this purpose, patients were selected from two databases (first, patients with T2DM; and second, patients with schizophrenia). In this way, the actual number of all patients with T2DM and concomitant schizophrenia was obtained for each year of observation separately. The data originated from all public health centres in Poland. In total, the study included 1,481,642 patients with schizophrenia, who were also diagnosed with T2DM. On average, 185,205 patients were verified for the presence of T2DM each year.

## Methods

**Type of study.** This study used a population-based observational framework [11]. This type of observational study includes all patients within a given jurisdiction, and is therefore less prone to the selection and referral biases that plague more traditional forms of observational research. Moreover, a population-based observational study framework has good external validity and provides insights into the delivery of care in routine practice to all patients, including the elderly.

**Source of database.** Information reported by healthcare entities to the tax payer in Poland (NFZ), on account of the health benefits provided to patients, was used for the retrospective data analysis for the period 2010–2017 (based on the existing provisions of the law [8]). Demographic data were collected from the web pages of Statistics Poland (GUS). The NFZ collects medical data from national and private medical centres that have signed a contract with NFZ and where medical services are financed from public funds. It should be noted that the collection of medical data by the NFZ is regulated by specific legal acts–the Decree of the Minister of Health of 20th June 2008 on the scope of the necessary information collected by public benefit providers, the detailed manner of registering this information, and its transfer to entities authorised to finance health benefits under public funding, which, together with its amending acts, remain in force (the most recent dated 1st July 2017). Data processing and coding were performed by the NFZ. For this reason, access to more detailed data was not available. This did not allow the analysis of additional risk factors, or more in-depth examination of medical data and patient characteristics. This is due to legal regulations and the maintenance of full anonymity.

**Ethical considerations.** This study is a retrospective data analysis from the period 2010–2017, using data from a public database. For this reason, approval from an independent ethics committee (IEC) was not necessary. The authors sought advice from the Bioethics Committee of the Medical University of Warsaw to conduct the present study. As the "commission does not issue opinions on survey, retrospective and other non-invasive scientific studies," approval was not required. Data owners have given their permission to use their data. Data processing, analysis of patient records and coding were performed by the NFZ. The data were fully

anonymised before we accessed them. For this reason, the IEC waived the requirement for informed consent.

**Statistical analyses.** Since the aggregated data were provided by the NFZ, only descriptive statistics were possible to present. The demographic characteristics of the participants were analysed through descriptive statistics. The annual prevalence of schizophrenia was estimated according to the T2DM diagnosis status, and the age groups were organised into eight sets (11–20, 21–30, 31–40, 41–50, 51–60, 61–70, 71 80 and ≥81). The eight sets were defined by the authors on the basis of data from the NFZ.

For the incidence of T2DM in Polish schizophrenic patients with reference to all patients with schizophrenia in the years 2010–2017, the relative risk (RR) with 95% confidence interval (95% CI) was calculated [12]. RR compares the risk of a health event among one group with the risk among another group. This calculation is the result of dividing the risk (occurrence of schizophrenia) in group 1 by the risk in group 2. These two groups differ in terms of the incidence of T2DM (e.g. T2DM patient population versus the entire Polish population). In calculating the RR, the following data were used: 1) the number of patients with T2DM over the age of 20 who sought treatment through the NFZ in the years 2010–2017; 2) the number of patients with schizophrenia over the age of 20 who sought treatment through the NFZ in the years 2010–2017; 3) the number of patients with T2DM and schizophrenia over the age of 20 who sought treatment through the NFZ in the years 2010–2017; and 4) the total number of Polish people over the age of 20 in the years 2010–2017. These data were obtained from the Central Statistical Office.

## Results

### Schizophrenia (F20 in total) and T2DM (E11) in Polish patients between 2010–2017 in state medical care

The conducted evaluation of the prevalence of schizophrenia (all subtypes) and T2DM in the entire population of patients with schizophrenia indicates that the trend of coexistence of the two illnesses was relatively stable in the years 2010–2017 and remained around the level of 13%. The coexistence rates for these two disorders were: 13.2% in 2010; 13.3% in 2011; 12.9% in 2012; 13.2% in 2013; 13.6% in 2014; 13.6% in 2015; 13.6% in 2016; and 13.5% in 2017.

In terms of sex, analysis shows that women prevail in the analysed group. In the case of women, the coexistence rates for these two disorders were: 8.7% in 2010; 8.8% in 2011; 8.5% in 2012; 8.6% in 2013; 8.9% in 2014; 8.8% in 2015; 8.8% in 2016; and 8.7% in 2017. For men, the coexistence rates were: 4.4% in 2010; 4.5% in 2011; 4.4% in 2012; 4.5% in 2013; 4.8% in 2014; 4.8% in 2015; 4.9% in 2016; and 4.8% in 2017. Comparing epidemiological data from 2010 with data from 2017, however, it can be noted that the number of men with T2DM and concomitant schizophrenia increased slightly, however women still constitute a significant percentage of the patients.

In terms of age, the prevalence of schizophrenia in T2DM in patients who sought treatment through the NFZ in the years 2010–2017 was varied and present in each age range, although to differing degrees. It should be noted that age groups with the greatest number of patients in state medical care with schizophrenia and T2DM do not overlap with the age of the patients with diagnosed T2DM. The greatest share of diagnosed cases of schizophrenia in patients with diabetes is in the age range 31–40 (around 14–16%) and was found to decrease with age. This correlation was noted in each analysed year (Fig 1).

### Relative risk of T2DM in schizophrenia

All subtypes of schizophrenia were analysed in total when calculating the relative risk. The analyses showed that the risk of T2DM in patients with schizophrenia is eight to ten times

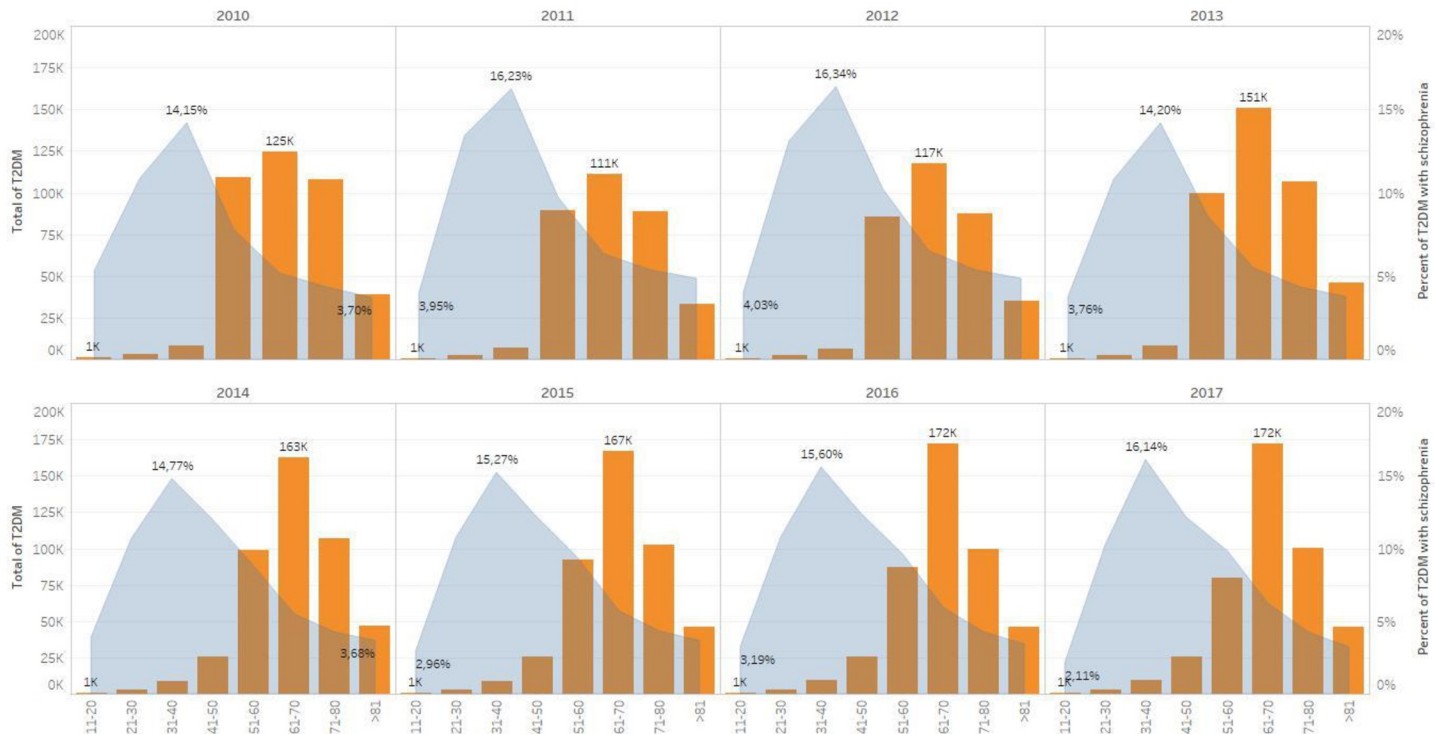

**Fig 1. Schizophrenia with concomitant T2DM, according to patient age.** The graph illustrates the percentage of diagnosed schizophrenia occurring in the total patient population with T2DM. On the horizontal axis, different age ranges are presented. The vertical axis indicates the percentage of patients with schizophrenia and T2DM in each age group. The age groups are divided into eight sets (11–20, 21–30, 31–40, 41–50, 51–60, 61–70, 71–80 and ≥81). For each year, the highest percentage of coexistence of schizophrenia and T2DM is shown with an indication of which age group it related to.

higher than in the general population. The RR of incidence of T2DM in schizophrenia was 8.33 (95% CI: 8.23–8.43) in 2017. RR was similar in all years of observation (Table 1). It should be noted that RR refers to patients who sought treatment through the NFZ in the years 2010–2017. This does not apply to the total number of patients.

## The frequency of coexistence of T2DM and subtypes of schizophrenia among men and women in the years 2010–2017

A detailed analysis of the frequency of coexistence of T2DM in various subtypes of schizophrenia, depending on sex, was also conducted. For paranoid schizophrenia (F20.0), there were

**Table 1. Relative risk (RR) of incidence of T2DM in schizophrenia.**

| Year | RR | 95% CI |
|---|---|---|
| 2010 | 8.24 | 8.14–8.34 |
| 2011 | 9.99 | 9.87–10.11 |
| 2012 | 9.67 | 9.55–9.79 |
| 2013 | 8.04 | 7.95–8.14 |
| 2014 | 8.04 | 7.95–8.14 |
| 2015 | 8.16 | 8.06–8.25 |
| 2016 | 8.26 | 8.16–8.36 |
| 2017 | 8.33 | 8.23–8.43 |

RR–relative risk; CI–confidence interval.

**Table 2. Rate of diagnosis of selected subtypes of schizophrenia among women and men with diagnosed T2DM (E11) in the years 2010–2017 in state medical care.**

| | | 2010 | 2011 | 2012 | 2013 | 2014 | 2015 | 2016 | 2017 |
|---|---|---|---|---|---|---|---|---|---|
| F20.0 | All | 126 | 114 | 93 | 74 | 87 | 112 | 62 | 83 |
| | Women | 78 (61.9%) | 66 (57.9%) | 66 (71.0%) | 55 (74.3%) | 58 (66.7%) | 74 (66.1%) | 37 (59.7%) | 57 (68.7%) |
| | Men | 48 (38.1%) | 48 (42.1%) | 27 (29.0%) | 19 (25.7%) | 29 (33.3%) | 38 (33.9%) | 25 (40.3%) | 26 (31.3%) |
| F20.1 | All | 56 | 40 | 43 | 48 | 53 | 67 | 52 | 49 |
| | Women | 38 (67.9%) | 24 (60.0%) | 23 (53.5%) | 28 (58.3%) | 31 (58.5%) | 43 (64.2%) | 37 (71.2%) | 33 (67.3%) |
| | Men | 18 (32.1%) | 16 (40.0%) | 20 (46.5%) | 20 (41.7%) | 22 (41.5%) | 24 (35.8%) | 15 (28.8%) | 16 (32.7%) |
| F20.2 | All | 109 | 137 | 147 | 147 | 198 | 239 | 240 | 217 |
| | Women | 75 (68.8%) | 95 (69.3%) | 106 (72.1%) | 97 (66.0%) | 135 (68.2%) | 169 (70.7%) | 165 (68.8%) | 141 (65.0%) |
| | Men | 34 (31.2%) | 42 (30.7%) | 41 (27.9%) | 50 (34.0%) | 63 (31.8%) | 70 (29.3%) | 75 (31.3%) | 76 (35.0%) |
| F20.3 | All | 45 | 67 | 75 | 77 | 80 | 113 | 109 | 125 |
| | Women | 26 (57.8%) | 43 (64.2%) | 52 (69.3%) | 50 (64.9%) | 54 (67.5%) | 87 (77.0%) | 82 (75.2%) | 90 (72.0%) |
| | Men | 19 (42.2%) | 24 (35.8%) | 23 (30.7%) | 27 (35.1%) | 26 (32.5%) | 26 (23.0%) | 27 (24.8%) | 35 (28.0%) |
| F20.4 | All | 21 | 59 | 76 | 105 | 112 | 139 | 134 | 144 |
| | Women | 15 (71.4%) | 37 (62.7%) | 45 (59.2%) | 58 (55.2%) | 63 (56.3%) | 87 (62.6%) | 84 (62.7%) | 87 (60.4%) |
| | Men | 6 (28.6%) | 22 (37.3%) | 31 (40.8%) | 47 (44.8%) | 49 (43.8%) | 52 (37.4%) | 50 (37.3%) | 57 (39.6%) |
| F20.5 | All | 251 | 369 | 446 | 499 | 477 | 501 | 526 | 549 |
| | Women | 148 (59.0%) | 213 (57.7%) | 267 (59.9%) | 299 (59.9%) | 283 (59.3%) | 284 (56.7%) | 303 (57.6%) | 326 (59.4%) |
| | Men | 103 (41.0%) | 156 (42.3%) | 179 (40.1%) | 200 (40.1%) | 194 (40.7%) | 217 (43.3%) | 223 (42.4%) | 223 (40.6%) |
| F20.6 | All | 115 | 268 | 309 | 310 | 355 | 462 | 456 | 514 |
| | Women | 72 (62.6%) | 157 (58.6%) | 177 (57.3%) | 189 (61.0%) | 220 (62.0%) | 303 (65.6%) | 288 (63.2%) | 315 (61.3%) |
| | Men | 43 (37.4%) | 111 (41.4%) | 132 (42.7%) | 121 (39.0%) | 135 (38.0%) | 159 (34.4%) | 168 (36.8%) | 199 (38.7%) |

F20.0 –paranoid schizophrenia; F20.1 –hebephrenic schizophrenia; F20.2 –catatonic schizophrenia; F20.3 –undifferentiated schizophrenia; F20.4 –post-schizophrenic depression; F20.5 –residual schizophrenia; F20.6 –simple schizophrenia.

751 cases reported over the eight years of follow-up study. On average, there were 94 patients with F20.0 and T2DM per year. For hebephrenic schizophrenia (F20.1), 408 cases were reported over the eight years of follow-up study. On average, there were 51 patients with F20.1 and T2DM. In the case of catatonic schizophrenia (F20.2), there were 1,434 reported cases over the eight years of follow-up study. On average, there were 179 patients with F20.2 and T2DM. For undifferentiated schizophrenia (F20.3), 691 cases were reported over the eight years of follow-up study. On average, there were 86 patients with F20.3 and T2DM. In the case of post-schizophrenic depression (F20.4), 790 cases were reported over the eight years of observation. On average, there were 99 patients with F20.4 and T2DM. For residual schizophrenia (F20.5), there were 3,618 reported cases over the eight years of follow-up study. On average, there were 452 patients with F20.5 and T2DM. For simple schizophrenia (F20.6), 2,789 cases were reported over the eight years of follow-up study. On average, there were 349 patients with F20.6 and T2DM. In each of the three subtypes of schizophrenia–paranoid schizophrenia (F20.0), hebephrenic schizophrenia (F20.1) and residual schizophrenia (F20.5)–women prevailed (Table 2). In Table 2, the total number of patients (N) refers to the specific subtypes of schizophrenia, not the total number of patients with schizophrenia who sought treatment through the NFZ in the years 2010–2017.

## The prevalence of T2DM in the selected subtypes of schizophrenia

A detailed analysis of the prevalence of the selected subtypes of schizophrenia in patients with T2DM indicate that paranoid schizophrenia (F20.0) was characterised by a relatively stable,

although subtly downward trend. As regards the prevalence of T2DM in hebephrenic schizophrenia (F20.1), catatonic schizophrenia (F20.2), undifferentiated schizophrenia (F20.3), post-schizophrenic depression (F20.4), residual schizophrenia (F20.5) and simple schizophrenia (F20.6), a rising trend was recorded (Fig 2).

## Discussion

This study, involving the analysis of the coexistence of schizophrenia (all subtypes analysed together) and T2DM in the period 2010 to 2017, indicate that the coexistence of the two illnesses remained relatively stable and T2DM concerned around 12.5–13% of patients with schizophrenia. The obtained results comply with other reports [4, 5, 7]. The present study is one of the few observational studies analysing the prevalence of diabetes in patients with schizophrenia. A similar study was conducted by a Danish team [13]. In the Danish study, however, incidence of diabetes was defined as prescription redemptions of insulin or oral anti-diabetic drugs, and ICD-10 diagnoses E10 or E11, while in the present study only diagnosis of E11 was analysed. Research has shown that patients with T2DM and schizophrenia have a higher percentage of complications observed compared to patients with T2DM without schizophrenia [13]. Similar observations have been made by Wu et al. [6] in Taiwan.

It should be noted that the presented results include an analysis of all cases of schizophrenia diagnosed in Poland in 2010–2017. For eight years of observation, 1,481,642 patients with schizophrenia were included, of which 185,205 patients were also diagnosed with T2DM. The collected medical data originated from national and private medical centres that had signed a contract with NFZ, where medical services were publicly funded. As a result, the obtained number of patients is representative of the entire population in Poland. Thus, it allows for a realistic estimation of the coexistence of these two disorders.

The present study analysed different types of schizophrenia with concomitant T2DM. In the literature, there are no such papers that have been able to accurately estimate the occurrence of T2DM in various subtypes of schizophrenia. Many studies are limited to reporting the coexistence of T2DM with schizophrenia, without dividing it by subtype. This may be related to diagnostic difficulties and in the literature, it is stressed that the diagnoses of these forms of disease may be made too arbitrarily according to subjective evaluation by doctors, and may therefore be insufficiently reliable and accurate [14].

In this study, the diagnosis of schizophrenia was based on the ICD-10 classification and was made by a physician. Our research shows that T2DM was present in all subtypes of schizophrenia, however the incidence of these two disorders was variable. Therefore, further studies assessing the coexistence of schizophrenia and T2DM should also take into account the subtypes of schizophrenia. From the data obtained, it can be seen that in most of the analysed subtypes of schizophrenia, the incidence of T2DM coexistence increased, but there are also situations in which the coexistence of these disorders was relatively constant. For example, there was a rising trend in the prevalence of T2DM in patients with residual schizophrenia (F20.5). It should be noted that negative symptoms are among the characteristic features of this subtype of schizophrenia, which may be connected with the gradual withdrawal from existing activities (e.g. professional, school or hobbies), as well as interpersonal relations. This may lead to total loss of interest and social withdrawal, and even avoidance of other people. An important accompaniment of the illness is also emotional distancing [9]. Additionally, it should be noted that residual schizophrenia is an advanced form of schizophrenia [9]; therefore, the symptoms of the disease and the reaction to antipsychotic drugs may significantly increase the risk of complications connected with the development of diabetes [15]. It is not clear whether this relationship may be a key factor in the coexistence of F20.5 and T2DM, and

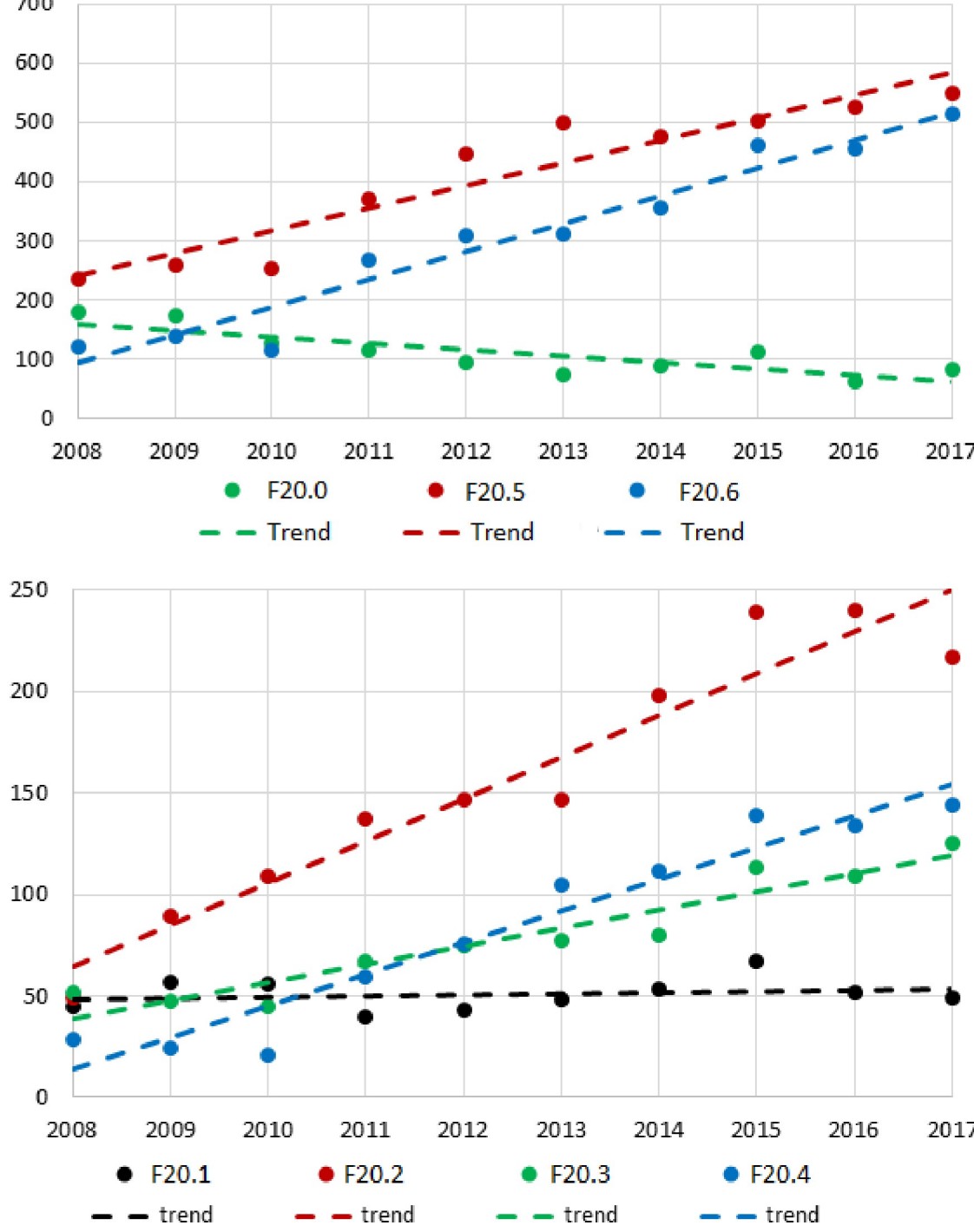

**Fig 2. Incidence of subtypes of schizophrenia in consecutive years, together with trend line of coexistence with T2DM.** Points represent the number of patients with different subtypes of schizophrenia. The dashed line indicates the trend of T2DM coexistence in different subtypes of schizophrenia. F20.0 –paranoid schizophrenia; F20.1 –hebephrenic schizophrenia; F20.2 –catatonic schizophrenia, F20.3 –undifferentiated schizophrenia; F20.4 –post-schizophrenic depression; F20.5 –residual schizophrenia F20.6 – simple schizophrenia.

as such requires further research. It should be borne in mind, however, that F20.5 is diagnosed when a person has a past history of at least one episode of schizophrenia, and relapse may occur after some time. Therefore, it is crucial to verify the risk factors for the recurrence of the disease in coexisting T2DM.

It is possible that changes in lifestyle and medical adherence may exist in patients with simple schizophrenia (F20.6) with T2DM, which may contribute to the worsening of the symptoms associated with T2DM in these patients. For example, Hjorth et al. [5] stress that patients newly diagnosed with schizophrenia exhibited high consumption of soft drinks and low physical activity. Together with the development of disease symptoms, it was observed that the physical profile of these patients worsened, with increased weight, waist circumference, visceral adiposity index and average glycosylated haemoglobin (HbA1c). Interestingly, observed correlations are more pronounced in men than in women, which should prompt further research regarding risk factor analysis.

Analysing data from the years 2010–2017, it was observed that the coexistence of T2DM with diagnosed paranoid schizophrenia remained relatively stable with a marginal downward trend. In line with ICD-10 [9], paranoid schizophrenia is a subtype of schizophrenia characterised by prominent delusions (typically persecutory or grandiose) or hallucinations, in the context of the relative preservation of cognitive function and affect. The prevalence of such symptoms may negatively influence activities connected with medical treatment. Research has shown that positive symptoms in schizophrenia (e.g. hallucinations and delusions) may significantly hamper adherence to medical recommendations. Additionally, medical adherence may be negatively affected by symptoms specific to schizophrenia, e.g. impairment of cognitive function, and the lack of adequate knowledge about the disease and methods of treatment [16]. Non-compliance with medical recommendations has a negative influence on the course of the illness, which may result in recurrence, further hospitalisation, longer periods before remission and suicide attempts [17].

The present study showed that the coexistence of T2DM and post-schizophrenic depression (F20.4) showed a slight upward trend. The coexistence of depression and T2DM has already been emphasised in the literature [7]. There are no studies, however, that analyse the coexistence of F20.4 with T2DM. In connection with the present observations, attention should be paid to the reasons for the coexistence of F20.4 with T2DM, which is also associated with numerous negative symptoms [9], possibly influencing the effectiveness of T2DM treatment. A relatively stable trend of coexistence of T2DM has also been noted in relation to patients who were diagnosed with hebephrenic schizophrenia. This subtype of schizophrenia is characterised by chaotic and inconsistent behaviour, and disorders in (inadequacy of) emotional reactions or lack thereof [9]. Limited understanding of the patient's behaviour may lead to difficulties in building a therapeutic alliance and a poor clinician-patient relationship, both of which have been shown to be significant predictors of non-adherence [16, 18]. These observations are particularly important in the treatment of patients with schizophrenia, whose behaviours may be incomprehensible for a doctor.

The presented data showed that these two diseases occurred more frequently in women than in men. These results stand in opposition to reports in the literature [19–21]. Given these differences, identifying the reasons why the coexistence of T2DM and schizophrenia may be more frequent in women than in men seems critical. Determining whether the observed correlation is actual, or rather stems from the fact that men with diagnosed schizophrenia less frequently undergo medical treatment, is also valid. The difference may stem from the fact that gender differences exist in the context of the initial effectiveness of treatment [21]. Thara & Kamath [21] stress that women seek medical advice more often and become more readily involved in treatment. Men may find it more difficult to cope with their own illness [21] and

may not look for medical advice. Impaired cognitive functioning (e.g. memory, concentration, attention, abstract thinking) in men with diagnosed schizophrenia and concomitant T2DM may be of key importance in this context as well. Men who do not undertake effective forms of T2DM treatment have higher levels of HbA1c and lipid disorders. Additionally, the development of schizophrenia itself negatively influences cognitive processes in men, which are intensified by diabetes, especially those regarding attention [22]. Considering that the obtained data demonstrates the more frequent prevalence of T2DM and schizophrenia in men, conducting further research within this scope is justified. The results, however, may also suggest that there is great need to plan for gender-sensitive mental health services, especially in the case of patients with T2DM and concomitant schizophrenia.

As regards age, it has been noted that the largest percentage of patients with schizophrenia with concomitant T2DM were between 31 and 40 years. In this regard, this age group merits particular attention. The obtained results are interesting in the context of studies concentrating on the age at which the first symptoms of schizophrenia were diagnosed. It should be noted that despite the fact that schizophrenia is diagnosed in both women and men with similar frequency, the onset of the illness typically occurs 3–5 years later in women than in men. Currently, it is assumed that the peak of morbidity with schizophrenia is between 21 and 25 years of age in men. In case of women, there are two peaks; one is between 25 and 30, and the other 40 to 50. This second peak in the development of schizophrenia found in women, represents 66–87% of patients, where the onset of the illness occurs between 40–50 years of age [23].

Taking into account the obtained results, it may be assumed that diabetes tends to occur in patients with schizophrenia after the age of peak morbidity with schizophrenia, namely after the age of 30. This would also confirm the observations where patients with diagnosed schizophrenia were noted to be at greater risk of developing diabetes [7], which was also demonstrated in our own study. Another important factor justifying the appearance of the peak of T2DM in the course of schizophrenia just after 30 years of age is treatment with antipsychotic medication. These drugs may lead to weight gain and greater risk of complications connected with obesity, including diabetes. This group of drugs may also directly influence glucose metabolism [15].

## Strengths and limitations

An important strength of this study is that all the data were retrieved from high quality Polish registers with almost 100% nationwide coverage. In addition, the number of patients included is large enough to guarantee reliable epidemiological estimates. The retrospective study design and the completeness of the follow-up of patients are further strengths of this study. Moreover, we adjusted for confounding factors, including age and sex. A final strong point of the present research is its eight year long duration. One of the most important limitations of the present research is its reliance on patients receiving state medical care, excluding privately treated patients. This is particularly applicable to patients with diagnosed schizophrenia. It is worth remembering that patients with schizophrenia represent around one fourth of patients in psychiatric hospitals. A large majority of the patients, however, are unaware of their symptoms and never consult a psychiatrist [24]. Another important limitation of the study is that it only takes into account patients who consciously sought medical advice, which means that they were invested in their own treatment. The number of patients who do not want to undergo medical treatment, or who are not aware of their condition, remains unknown.

## Conclusion

This study demonstrates that the prevalence of T2DM in schizophrenia over the years 2010–2017 was relatively stable, and the largest percentage of patients were noted to be in the age

range 31–40 years. In this regard, taking action to detect diabetes early in patients with schizophrenia is justified, although these actions should be gender-dependent. There remains a great need to plan for gender-sensitive mental health services, as well as to take adequate action aimed at boosting the effectiveness of diabetological care among patients with schizophrenia, and their adherence to medical recommendations.

## Supporting information

**S1 File.**
(PDF)

**S1 Data.**
(XLSX)

## Author Contributions

**Conceptualization:** Mariusz Jaworski, Mariusz Panczyk, Joanna Gotlib.

**Data curation:** Mariusz Jaworski, Andrzej Śliwczyński.

**Formal analysis:** Mariusz Jaworski, Mariusz Panczyk, Melania Brzozowska, Joanna Gotlib.

**Methodology:** Mariusz Jaworski, Mariusz Panczyk, Andrzej Śliwczyński, Melania Brzozowska, Joanna Gotlib.

**Project administration:** Mariusz Jaworski.

**Resources:** Andrzej Śliwczyński.

**Supervision:** Mariusz Panczyk, Andrzej Śliwczyński, Melania Brzozowska.

**Writing – original draft:** Mariusz Jaworski, Mariusz Panczyk, Andrzej Śliwczyński, Melania Brzozowska, Joanna Gotlib.

**Writing – review & editing:** Mariusz Jaworski, Mariusz Panczyk, Joanna Gotlib.

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
