## [Decision Letter · Decision Letter 0]

30 Jun 2020

PONE-D-20-07489

SCHIZOPHRENIC PATIENT AT A DIABETOLOGIST'S: AN 8-YEAR POPULATION BASED OBSERVATIONAL STUDY

PLOS ONE

Dear Dr. Jaworski,

Thank you for submitting your manuscript to PLOS ONE. After careful consideration, we feel that it has merit but does not fully meet PLOS ONE’s publication criteria as it currently stands. Therefore, we invite you to submit a revised version of the manuscript that addresses the points raised during the review process.

We look forward to receiving your revised manuscript.

Kind regards,

Zhenghui Yi

Academic Editor

PLOS ONE

Journal Requirements:

2. In your ethics statement in the manuscript and in the online submission form, please provide additional information about the patient records used in your retrospective study. Specifically, please ensure that you have discussed whether all data were fully anonymized before you accessed them and/or whether the IRB or ethics committee waived the requirement for informed consent. If patients provided informed written consent to have data from their medical records used in research, please include this information.'"

3. We noticed you still have some minor occurrence of overlapping text with previous publications (https://doi.org/10.3109/07435800.2016.1141948), which needs to be addressed. In your revision ensure you cite all your sources, and quote or rephrase any duplicated text outside the methods section. Further consideration is dependent on these concerns being addressed

Additional Editor Comments (if provided):

The format of the results section does not meet the standards of the journal. The Figure1, Figure2, Figure3 etc. between paragraphs should be removed. And in the results section, detailed data and results should be presented by words in text, rather than just saying “Detailed data are presented in Figure… or table…”. I also have a suggestion that the language of the manuscript needs to be improved.

Reviewers' comments:

Reviewer's Responses to Questions

**Comments to the Author**

1. Is the manuscript technically sound, and do the data support the conclusions?

Reviewer #1: Partly

2. Has the statistical analysis been performed appropriately and rigorously? 

Reviewer #1: N/A

3. Have the authors made all data underlying the findings in their manuscript fully available?

Reviewer #1: Yes

4. Is the manuscript presented in an intelligible fashion and written in standard English?

Reviewer #1: No

5. Review Comments to the Author

Reviewer #1: In this study the authors examined the prevalence of T2DM among Polish schizophrenia patients. There were several positives to this study. However, there were also some negatives to this study, as detailed below. The major concern with the manuscript is that its currently new finding’s contributions is relatively limited to this area. The authors should add more new findings in addition to what is already known about this subject, for example, what are the associated risk factors in this sample?

1) The authors used data from NFZ in the years from 2010-2017, that is a bit old, since we are in the middle of 2020 now.

2) The authors should report sample size in the paper (abstract, methods and results).

3) The first paragraph of the introduction is too long, and it should be summarized into two or three sentences.

4) The hypotheses at the end of the introduction are lacking in specificity and insufficiently motivated. And, The ICD-10 code for schizophrenia is F20, which includes paranoid schizophrenia, hebephrenic schizophrenia, catatonic schizophrenia, undifferentiated schizophrenia, postschizophrenic depression, residual schizophrenia, simple schizophrenia, other schizophrenia, and schizophrenia, unspecified. A rational for the use of three subtypes here should be provided.

5) There is no need to present such limited information in two figures, i.e., Fig1 and Fig2, one figure would work fine.

6) Fig3 is not clear, please make it clear when resubmission.

7) Detailed figure legends should be provided.

8) The authors should provide a table with detailed demographic information and clinical variables about the sample.

9) The discussion is too long, and it should be discussed in more depth.

10) Could you please kindly provide the pdf of reference 14 [Nielsen AT, 2018]? Thanks!

11) Lead your limitations section with the strengths of your study!

12) The manuscript includes some non-standard phrasings and could use editing for language prior to resubmission.

6. PLOS authors have the option to publish the peer review history of their article (what does this mean?). If published, this will include your full peer review and any attached files.

Reviewer #1: No

---

## [Author Response · Author response to Decision Letter 0]

18 Aug 2020

Dear Editor of PLOS ONE 

Thank you very much for all your comments and the opportunity to improve the manuscript. We have been working to introduce the comments provided by Reviewers. We hope that the present version of the manuscript will meet the expectations of the Reviewers and Editor of the PLOS ONE. The changes within the revised manuscript were highlighted (underlined and in yellow (#R1- Reviewer and #E -Editor). Once again, thank you for your valuable comments and attention given to our paper.

Best regards,

Mariusz Jaworski 

 Reviewer (#R) Comments Authors' Response (#A)

#1 #R

The major concern with the manuscript is that its currently new finding’s contributions is relatively limited to this area. The authors should add more new findings in addition to what is already known about this subject, for example, what are the associated risk factors in this sample? #A

Thank you very much for this comment. For us, it is important to identify which sub-types of schizophrenia may bear serious consequences in the context of patient treatment with T2DM. There is no such data available at the moment. This is important because the rate of co-occurrence of schizophrenia and T2DM cited in the literature is too vague. Moreover, this indicator does not take into account the prevalence of T2DM in different types of schizophrenia. For this reason, in this paper, we wanted to determine if T2DM is present in every sub-type of schizophrenia. 

See added information (yellow):

Page 2, lines 9-10

In abstract section:

‘The incidence of T2DM has been assessed in various sub-types of schizophrenia’

In introduction section

Page 3, lines 27-28

‘It is difficult to determine the exact frequency of the coexistence of schizophrenia and T2DM. The results vary widely’

Page 4, lines 1-10

‘Any large discrepancies may be related to methodological problems: for example, sample size estimation; methods of diagnosing T2DM and schizophrenia; the form of research (cross-sectional or longitudinal studies); and the criteria for inclusion in the study (such as age and sex). Therefore, it is important to select the appropriate sample size, which would take into account not only the appropriate number of patients, but also their age and sex characteristics (4). For this reason, it may be helpful to analyse large databases that include the total number of patients receiving medical care. Such databases include patients with medical diagnoses from all over the country. By using such databases, it is possible to estimate the actual co-occurrence of type 2 diabetes and schizophrenia, thus, obtaining more realistic indicators, as well as minimising the risk of measurement error to a minimum’.

See: Page 4, lines 28-31 and page 5, lines 1-4

‘Bearing in mind, however, that schizophrenia is not a homogeneous illness, special attention has been paid to all sub-types: paranoid schizophrenia (F20.0), hebephrenic schizophrenia (F20.1), catatonic schizophrenia (F20.2), undifferentiated schizophrenia (F20.3), post-schizophrenic depression (F20.4), residual schizophrenia (F20.5) and simple schizophrenia (F20.6) [9]. It is important to identify which sub-types of schizophrenia may bear serious consequences in the context of patient treatment with T2DM. In this paper, we wanted to determine if T2DM is present in every sub-type of schizophrenia. There is no such data available at the moment’

In result section:

See page 12, lines 8-11

‘As regards the prevalence of T2DM in hebephrenic schizophrenia (F20.1), catatonic schizophrenia (F20.2), undifferentiated schizophrenia (F20.3), post-schizophrenic depression (F20.4), residual schizophrenia (F20.5) and simple schizophrenia (F20.6), a rising trend was recorded (Fig 2).

#2 #R

The authors used data from NFZ in the years from 2010-2017, that is a bit old, since we are in the middle of 2020 now. #A

It is true that we used data from NFZ in the years from 2010-2017. It was related to several factors. Firstly, data processing and coding were performed by the NFZ. For this reason, access to more detailed data has been blocked. We could only use the data that the NFZ gave us.

Secondly, The Polish National Health Fund (NFZ) collects medical data from national and private medical centres that have signed a contract with NFZ, where medical services are financed from public funds. These reports are collected periodically. Before we have access to them, the data must be properly prepared. It takes time.

Third, properly preparing the script also takes time. In our case, It is important to identify which sub-types of schizophrenia may bear serious consequences in the context of patient treatment with T2DM. For this reason, we have decided to take data from 2010-2017. Of course, we will also take into account newer data from 2018-2019 in future publications.

#3 #R

The authors should report sample size in the paper (abstract, methods and results). #A

Sample size has been provided

In the Methods section, we added the Simple size section and added the necessary information. 

See added information (yellow):

Page 6, lines 4-6

‘The sample size included all patients registered in the NFZ database who used medical services financed from public funds’

In the Simple size section, we added the total number of T2DM and schizophrenia patients in each years of observation.

See added information (yellow):

See page 6, lines 27-31 and page 7, lines 1-7

In the second step, patients diagnosed with schizophrenia were selected. The sub-type of schizophrenia was also included. In particular years of observation, the following total numbers of patients with schizophrenia were selected: 80,807 (in 2010), 184,556 (in 2011), 189,683 (in 2012), 188,536 (in 2013), 187,151 (in 2014), 185,651 (in 2015), 183,587 (in 2016) and 181,716 (in 2017).

In the final step, patients with coexisting T2DM and schizophrenia were selected, regardless of sub-type. For this purpose, patients were selected from two databases (first, patients with T2DM; and second, patients with schizophrenia). In this way, the actual number of all patients with T2DM and concomitant schizophrenia was obtained for each year of observation separately. The data originated from all public health centres in Poland. In total, the study included 1,481,642 patients with schizophrenia, who were also diagnosed with T2DM. On average, 185,205 patients were verified for the presence of T2DM each year.

In abstract section, we added number of patients

See page 2, lines 11-12

‘In eight years of follow-up studies, 1,481,642 patients with schizophrenia were included, of which 185,205 were also diagnosed with T2DM’

#4 #R

The first paragraph of the introduction is too long, and it should be summarized into two or three sentences. #A

As suggested by the reviewer, this part of the introduction has been modified.

See modified content:

See page 3, lines 2-8

‘Schizophrenia is a severe mental illness (SMI), which is characterised by cognitive, emotional, perceptual and behavioural disorders (1, 2). The prevalence of schizophrenia is very similar in many countries and a lifetime risk of schizophrenia is around .2–1% (1) without differences between sexes. It is believed that schizophrenia is a disease that has serious consequences not only for one’s health (disability, comorbidities, complications), but also economic and social ones. In addition, it is emphasised that patients with schizophrenia receive fewer healthcare services compared to the general population (3)’

#5 #R

The hypotheses at the end of the introduction are lacking in specificity and insufficiently motivated. And, The ICD-10 code for schizophrenia is F20, which includes paranoid schizophrenia, hebephrenic schizophrenia, catatonic schizophrenia, undifferentiated schizophrenia, postschizophrenic depression, residual schizophrenia, simple schizophrenia, other schizophrenia, and schizophrenia, unspecified. A rational for the use of three subtypes here should be provided. #A

Thank you very much for this comment. To the article, we have added the missing data on other types of schizophrenia.

In ‘Introduction’ part, we also added information on the analysis of the prevalence of different types of schizophrenia in type 2 diabetes. This is the asset of this article, because the rate of co-occurrence of schizophrenia and T2DM cited in the literature is too vague. This indicator does not take into account the prevalence of T2DM in different types of schizophrenia. There is no such data available at the moment. 

Moreover, it is not known whether all sub-type of schizophrenia occur with the same frequency in patients with type 2 diabetes. Therefore, the aim of our article is not only to show the general rate of co-occurrence of schizophrenia and T2DM, but also to determine if T2DM is present in every sub-type of schizophrenia

See added information (yellow):

Page 4, lines 11-21

‘This is important because the rate of co-occurrence of schizophrenia and T2DM cited in the literature is too vague. Moreover, this indicator does not take into account the prevalence of T2DM in different types of schizophrenia. This indicator treats schizophrenia as a uniform disease, while the ICD-10 code for schizophrenia (F20) includes paranoid schizophrenia, hebephrenic schizophrenia, catatonic schizophrenia, undifferentiated schizophrenia, post-schizophrenic depression, residual schizophrenia and simple schizophrenia. Therefore, it is crucial to verify whether T2DM occurs with the same frequency in different sub-types of schizophrenia or not. Currently, it is difficult to determine which of the sub-types of schizophrenia may have a greater tendency to coexist with T2DM. Such detailed data will allow the development of more adequate health education methods, but also provide more reliable and credible data’.

#6 #R

There is no need to present such limited information in two figures, i.e., Fig1 and Fig2, one figure would work fine. #A

Figure 1 is deleted and the data is in the text.

See added text (yellow):

Page 9, lines 1-3

‘The coexistence rate for these two disorders was 13.2% in 2010; 13.3% in 2011; 12.9% in 2012; 13.2% in 2013; 13.6% in 2014; 13.6% in 2015; 13.6% in 2016; and 13.5% in 2017’.

Also the data presented in figure 2 is in the text. Figure 2 is deleted.

See added text (yellow):

Page 9, lines 4-8

‘In terms of sex, the analysis has shown that women prevail in the analysed group. In the case of women, the coexistence rate for these two disorders was 8.7% in 2010; 8.8% in 2011; 8.5% in 2012; 8.6% in 2013; 8.9% in 2014; 8.8% in 2015; 8.8% in 2016; and 8.7% in 2017. For men, the coexistence rate was 4.4% in 2010; 4.5% in 2011; 4.4% in 2012; 4.5% in 2013; 4.8% in 2014; 4.8% in 2015; 4.9% in 2016; and 4.8% in 2017’.

#7 #R

Fig3 is not clear, please make it clear when resubmission. #A

Done. Fig 3 is now Fig 1. 

#8 #R

Detailed figure legends should be provided. #A

Done. We added legends to figures 1 and 2.

See added information (yellow)

For Fig. 1 see page 9, lines 20-26

‘Legend: The graph illustrates the percentage of schizophrenia occurring in the total patient population with T2DM. On the horizontal line, different age ranges have been presented. The right vertical line indicates the percentage of patients with schizophrenia and T2DM in specific age groups. The age groups were divided into eight sets (11 to 20, 21 to 30, 31 to 40, 41 to 50, 51 to 60, 61 to 70, 71 to 80 and ≥81). For each year, the highest percentage of coexistence of F20 and T2DM was shown with an indication of which age group it related to’.

For Fig. 2 see page 12, lines 12-18 

‘Legend: In the graph, the points represent the number of patients with different sub-types of schizophrenia. The dashed line indicates the trend of T2DM coexistence in different sub-types of schizophrenia. F20.0 – paranoid schizophrenia; F20.1 – hebephrenic schizophrenia; F20.2 – catatonic schizophrenia, F20.3 – undifferentiated schizophrenia; F20.4 – post-schizophrenic depression; F20.5 – residual schizophrenia F20.6 – simple schizophrenia’.

We also added legends to tables 1 and 2.

For table 1 see page 10, line 7

For table 2 see page 12, lines 1-3

#9 #R

The authors should provide a table with detailed demographic information and clinical variables about the sample. #A

We cannot provide a table with this data because we do not have access to it. We do not have permission for this. It is related to legal regulations and the procedure of data extraction from the database. It was not possible to obtain such data. We have added this information in our article.

See added information (yellow):

See page 7, lines 27-31.

‘Data processing and coding were performed by the NFZ. For this reason, access to more detailed data has been blocked. This did not allow the analysis of additional risk factors, and more in-depth examination of medical data and patient characteristics. This is due to legal regulations and the maintenance of full anonymity’.

And page 8, lines 7-10

‘Data processing, analysing patient records and coding were performed by the NFZ. The data were fully anonymized before we accessed them. For this reason, the IEC waived the requirement for informed consent.

Page 4, lines 25-26

The database developed by the National Health Fund (NFZ) includes all patients using public healthcare in Poland.

#10 #R

The discussion is too long, and it should be discussed in more depth. #A

Thank you very much for this comment. As suggested by the reviewer, this part of the paper has been modified.

See modified part (yellow):

Page 12, lines 20-25

‘This study, involving the analysis of the coexistence of schizophrenia (all sub-types analysed together) and T2DM in the period 2010 to 2017, indicated that the coexistence of the two illnesses remains relatively stable and concerns around 12.5–13% of patients with schizophrenia. The obtained results comply with other reports (4, 5, 7). The present study is one of the few observational studies analysing the prevalence of diabetes in patients with schizophrenia. A similar study was conducted by a Danish team (13)’

Page 13, lines 4-11

‘It should be noted that the presented results include an analysis of all cases of schizophrenia diagnosed in Poland in 2010–2017. For eight years of observation, 1,481,642 patients with schizophrenia were included, of which 185,205 patients were also diagnosed with T2DM. The collected medical data originated in national and private medical centres that had signed a contract with NFZ, where medical services were publicly funded. As a result, the obtained number of patients is representative of the entire population in Poland. Thus, it allows for a realistic estimation of the problem of the coexistence of these two disorders’.

Page 13, lines 12-15

‘The present study analysed different types of schizophrenia with concomitant T2DM. In the literature, there are no such papers that would allow for an accurate estimation of the occurrence of T2DM in various sub-types of schizophrenia. Many studies are limited to reporting the coexistence of T2DM with schizophrenia, without dividing it by sub-type’

Page 13, lines 19-26

‘In the present study, the diagnosis of schizophrenia was based on the ICD-10 classification and was made by a physician. Our research shows that T2DM is present in all types of schizophrenia. The incidence of these two disorders, however, is variable. Therefore, further studies assessing the coexistence of schizophrenia and T2DM should also take into account the type of schizophrenia. From the data obtained, it can be seen that in most of the analysed types of schizophrenia, the incidence of T2DM coexistence increases, but there are also situations in which the coexistence of these disorders is relatively constant’.

Page 14, lines 5-9

‘Thus, it should be checked whether this relationship can be a key factor in the coexistence of F20.5 and T2DM. This is not clear and requires further research. It should be borne in mind, however, that F20.5 is diagnosed when a person has a past history of at least one episode of schizophrenia. Relapse occurs after some time. Therefore, it is crucial to verify the risk factors for the recurrence of the disease in coexisting T2DM.’

Page 14, lines 10-12

‘It is possible that changes in lifestyle and medical adherence may exist in patients with simple schizophrenia (F20.6) with T2DM, which may contribute to the worsening of the symptoms associated with T2DM in these patients’.

Page 15, lines 3-9

The present study showed that the coexistence of T2DM and post-schizophrenic depression (F20.4) shows a slight upward trend. The coexistence of depression and T2DM is emphasised in the literature (7). There are no studies, however, that analyse the coexistence of F20.4 with T2DM. In connection with the present observations, attention should be paid to the reasons for the coexistence of F20.4 with T2DM, which is also associated with numerous negative symptoms of F20.4 (8), possibly influencing the effectiveness of T2DM treatment.

#11 #R

Could you please kindly provide the pdf of reference 14 [Nielsen AT, 2018]? Thanks! #A

This source was published in Schizophrenia Bulletin as an abstract of a conference speech during Oral Session: Comorbidity. 

(the Sixth Biennial SIRS Conference). We added pdf

#12 #R

Lead your limitations section with the strengths of your study! #A

We've added strengths of our study in “limitation and strengths” section. 

See added text:

Page 16, lines 27-31and page 17, lines 1-2

‘An important strength of this study is that all the data were retrieved from high-quality Polish registers with almost 100% nationwide coverage. In addition, the number of patients included is large enough to guarantee reliable epidemiological estimates. The retrospective study design and the completeness of the follow-up of patients are among further major strengths of this study. Moreover, we adjusted for confounding factors, including age and sex. A final strong point of the present research is its eight-year long duration. 

#13 #R

The manuscript includes some non-standard phrasings and could use editing for language prior to resubmission. #A

The aforementioned article has been revised in terms of linguistic corrections that included spelling, punctuation, grammar, and usage mistakes. 

See added certificate of proofreading

 Editor (#E) Comments Authors' Response (#A)

#1 #E 

The format of the results section does not meet the standards of the journal. The Figure1, Figure2, Figure3 etc. between paragraphs should be removed. 

 #A

Done.

#2 #E 

And in the results section, detailed data and results should be presented by words in text, rather than just saying “Detailed data are presented in Figure… or table…”. 

 #A

Done.

Figure 1 is deleted and the data is in the text.

See added text (yellow):

Page 9, lines 1-3

‘The coexistence rate for these two disorders was 13.2% in 2010; 13.3% in 2011; 12.9% in 2012; 13.2% in 2013; 13.6% in 2014; 13.6% in 2015; 13.6% in 2016; and 13.5% in 2017’.

Also the data presented in figure 2 is in the text. Figure 2 is deleted.

See added text (yellow):

Page 9, lines 4-8

‘In terms of sex, the analysis has shown that women prevail in the analysed group. In the case of women, the coexistence rate for these two disorders was 8.7% in 2010; 8.8% in 2011; 8.5% in 2012; 8.6% in 2013; 8.9% in 2014; 8.8% in 2015; 8.8% in 2016; and 8.7% in 2017. For men, the coexistence rate was 4.4% in 2010; 4.5% in 2011; 4.4% in 2012; 4.5% in 2013; 4.8% in 2014; 4.8% in 2015; 4.9% in 2016; and 4.8% in 2017’.

In Relative risk of T2DM in schizophrenia section, we added some numbers.

See page 10, lines 1-3

‘The relative risk (RR) of incidence of T2DM in schizophrenia was 8.33 (8.23–8.43) in 2017. RR was very similar in all years of observation’

We also added some numbers in part related to The frequency of coexistence of T2DM and sub-types of schizophrenia.

See page 10, lines 8-9 and lines 11-24

‘The frequency of coexistence of T2DM and sub-types of schizophrenia among men and women in the years 2010–2017’

‘For paranoid schizophrenia (F20.0), there were 751 cases reported over eight years of follow-up studies. On average, there were 94 patients with F20.0 and T2DM. For hebephrenic schizophrenia (F20.1), 408 cases have been reported over eight years of follow-up studies. On average, there were 51 patients with F20.1 and T2DM. In the case of catatonic schizophrenia (F20.2), there were 1,434 reported cases over eight years of follow-up studies. On average, there were 179 patients with F20.2 and T2DM. For undifferentiated schizophrenia (F20.3), 691 cases have been reported over eight years of follow-up studies. On average, there were 86 patients with F20.3 and T2DM. In the case of post-schizophrenic depression (F20.4), 790 cases were reported over eight years of observation. On average, there were 99 patients with F20.4 and T2DM. For residual schizophrenia (F20.5), there were 3,618 reported cases over eight years of follow-up studies. On average, there were 452 patients with F20.5 and T2DM. For simple schizophrenia (F20.6), 2,789 cases have been reported over eight years of follow-up studies. On average, there were 349 patients with F20.6 and T2DM’.

#3 #E 

I also have a suggestion that the language of the manuscript needs to be improved. #A

The aforementioned article has been revised in terms of linguistic corrections that included spelling, punctuation, grammar, and usage mistakes. 

See added certificate of proofreading

#4 #E 

Please ensure that your manuscript meets PLOS ONE's style requirements, including those for file naming. #A

Done.

#5 #E 

In your ethics statement in the manuscript and in the online submission form, please provide additional information about the patient records used in your retrospective study. Specifically, please ensure that you have discussed whether all data were fully anonymized before you accessed them and/or whether the IRB or ethics committee waived the requirement for informed consent. If patients provided informed written consent to have data from their medical records used in research, please include this information.'"

 #A

We added additional information in Ethical considerations section.

See page 8, lines 7-10

‘Data processing, analysing patient records and coding were performed by the NFZ. The data were fully anonymized before we accessed them. For this reason, the IEC waived the requirement for informed consent.

#6 #E 

We noticed you still have some minor occurrence of overlapping text with previous publications (https://doi.org/10.3109/07435800.2016.1141948), which needs to be addressed. In your revision ensure you cite all your sources, and quote or rephrase any duplicated text outside the methods section. Further consideration is dependent on these concerns being addressed

 #A

I verified it using the plagiarism program. Everything is ok. If there is still a problem, please share the results of this study with me.

#7 #E 

In your Data Availability statement, you have not specified where the minimal data set underlying the results described in your manuscript can be found. PLOS defines a study's minimal data set as the underlying data used to reach the conclusions drawn in the manuscript and any additional data required to replicate the reported study findings in their entirety. All PLOS journals require that the minimal data set be made fully available. #A

All the data which we got from the NFZ you can find in the tables in our manuscript. We only got cumulative data.

---

## [Decision Letter · Decision Letter 1]

6 Jan 2021

PONE-D-20-07489R1

SCHIZOPHRENIC PATIENTS AT A DIABETOLOGIST'S: AN 8-YEAR POPULATION-BASED OBSERVATIONAL STUDY

PLOS ONE

Dear Dr. Jaworski,

Thank you for submitting your manuscript to PLOS ONE. After careful consideration, we feel that it has merit but does not fully meet PLOS ONE’s publication criteria as it currently stands. Therefore, we invite you to submit a revised version of the manuscript that addresses the points raised during the review process.

We look forward to receiving your revised manuscript.

Kind regards,

Muhammad Sajid Hamid Akash

Academic Editor

PLOS ONE

Reviewers' comments:

Reviewer's Responses to Questions

**Comments to the Author**

1. If the authors have adequately addressed your comments raised in a previous round of review and you feel that this manuscript is now acceptable for publication, you may indicate that here to bypass the “Comments to the Author” section, enter your conflict of interest statement in the “Confidential to Editor” section, and submit your "Accept" recommendation.

Reviewer #1: All comments have been addressed

Reviewer #2: All comments have been addressed

2. Is the manuscript technically sound, and do the data support the conclusions?

Reviewer #1: Yes

Reviewer #2: Yes

3. Has the statistical analysis been performed appropriately and rigorously? 

Reviewer #1: Yes

Reviewer #2: No

4. Have the authors made all data underlying the findings in their manuscript fully available?

Reviewer #1: Yes

Reviewer #2: (No Response)

5. Is the manuscript presented in an intelligible fashion and written in standard English?

Reviewer #1: Yes

Reviewer #2: No

6. Review Comments to the Author

Reviewer #1: In general, the authors have been responsive to my questions of review. The results are novel and interesting and the methodology seems correct. Overall, I think it is worthy of publication and will be of interest to readers.

Reviewer #2: How the data was analyzed. Which test was used to explain your hypothesis?? Lots of grammatical and typo mistakes e.g., In this paper, we wanted to determine if T2DM is present in every sub-type of schizophrenia??

7. PLOS authors have the option to publish the peer review history of their article (what does this mean?). If published, this will include your full peer review and any attached files.

Reviewer #1: No

Reviewer #2: No

---

## [Author Response · Author response to Decision Letter 1]

18 Feb 2021

Dear Editor of PLOS ONE 

Thank you very much for all your comments and the opportunity to improve the manuscript. We have been working to introduce the comments provided by Reviewers. We hope that the present version of the manuscript will meet the expectations of the Reviewers and Editor of the PLOS ONE. The changes within the revised manuscript were highlighted (underlined and in yellow). Once again, thank you for your valuable comments and attention given to our paper.

Best regards,

Reviewer (#R) Comments 

In general, the authors have been responsive to my questions of review. The results are novel and interesting and the methodology seems correct. Overall, I think it is worthy of publication and will be of interest to readers. 

Authors' Response (#A)

We would like to thank the reviewer for the positive review and recommendation of our article for publication in the PLOS ONE journal.

Reviewer (#R) Comments 

How the data was analyzed. Which test was used to explain your hypothesis??

Authors' Response (#A)

As the aggregated data was provided by the National Health Fund, only descriptive statistics could be presented. However, to estimate the population value of the relative risk (RR), interval estimation was used as a method of statistical inference. For the point estimator (RR) a 95% confidence interval was calculated. Due to the nature of the collected data, the null hypothesis significance test was not used

Reviewer (#R) Comments 

Lots of grammatical and typo mistakes e.g., In this paper, we wanted to determine if T2DM is present in every sub-type of schizophrenia??

Is the manuscript presented in an intelligible fashion and written in standard English? No #A

Thanks for this comment. Our article has been edited by a native speaker of English who is a Doctor of Medicine. I am attaching the certificate. We hope that this version of our article is linguistically much better.

Reviewer (#R) Comments 

Has the statistical analysis been performed appropriately and rigorously? No

Authors' Response (#A)

We have added information on the rules for calculating relative risk. Additional data that was used were also explained. Please see the modified and completed text: 

Since the aggregated data were provided by the NFZ, only descriptive statistics were possible to present. The demographic characteristics of the participants were analysed through descriptive statistics. The annual prevalence of schizophrenia was estimated according to the T2DM diagnosis status, and the age groups were organised into eight sets (11–20, 21–30, 31–40, 41–50, 51–60, 61–70, 71 80 and ≥81). The eight sets were defined by the authors on the basis of data from the NFZ. For the incidence of T2DM in Polish schizophrenic patients with reference to all patients with schizophrenia in the years 2010–2017, the relative risk (RR) with 95% confidence interval (95% CI) was calculated (12). RR compares the risk of a health event among one group with the risk among another group. This calculation is the result of dividing the risk (occurrence of schizophrenia) in group 1 by the risk in group 2. These two groups differ in terms of the incidence of T2DM (e.g. T2DM patient population versus the entire Polish population).

Page 8 section Statistical analyses

---

## [Editor Report · Decision Letter 2]

26 Feb 2021

SCHIZOPHRENIC PATIENTS WITH TYPE 2 DIABETES: AN 8-YEAR POPULATION-BASED OBSERWATIONAL STUDY

PONE-D-20-07489R2

Dear Dr. Jaworski,

We’re pleased to inform you that your manuscript has been judged scientifically suitable for publication and will be formally accepted for publication once it meets all outstanding technical requirements.

Kind regards,

Muhammad Sajid Hamid Akash

Academic Editor

PLOS ONE
---

## [Editor Report · Acceptance letter]

3 Mar 2021

PONE-D-20-07489R2 

SCHIZOPHRENIC PATIENTS WITH TYPE 2 DIABETES: AN 8-YEAR POPULATION-BASED OBSERVATIONAL STUDY 

Dear Dr. Jaworski:

I'm pleased to inform you that your manuscript has been deemed suitable for publication in PLOS ONE. Congratulations! Your manuscript is now with our production department. 

Kind regards, 

on behalf of

Dr. Muhammad Sajid Hamid Akash 

Academic Editor

PLOS ONE